# Cataleptogenic Effect of Haloperidol Formulated in Water-Soluble Calixarene-Based Nanoparticles

**DOI:** 10.3390/pharmaceutics15030921

**Published:** 2023-03-11

**Authors:** Nadezda E. Kashapova, Ruslan R. Kashapov, Albina Y. Ziganshina, Dmitry O. Nikitin, Irina I. Semina, Vadim V. Salnikov, Vitaliy V. Khutoryanskiy, Rouslan I. Moustafine, Lucia Y. Zakharova

**Affiliations:** 1Arbuzov Institute of Organic and Physical Chemistry, FRC Kazan Scientific Center of RAS, 8 Arbuzov Str., 420088 Kazan, Russia; 2Department of Pharmacology, Kazan State Medical University, 49 Butlerov Str., 420012 Kazan, Russia; 3Kazan Institute of Biochemistry and Biophysics, FRC Kazan Scientific Center of RAS, 2/31 Lobachevsky Str., 420111 Kazan, Russia; 4School of Pharmacy, University of Reading, Whiteknights, Reading RG6 6DX, UK; 5Institute of Pharmacy, Kazan State Medical University, 16 Fatykh Amirkhan Str., 420126 Kazan, Russia

**Keywords:** calixarene, haloperidol, mucoadhesion, self-assembly, TEM, nanoparticles, toxicity in vivo, open field, catalepsy

## Abstract

In this study, a water-soluble form of haloperidol was obtained by coaggregation with calix[4]resorcinol bearing viologen groups on the upper rim and decyl chains on the lower rim to form vesicular nanoparticles. The formation of nanoparticles is achieved by the spontaneous loading of haloperidol into the hydrophobic domains of aggregates based on this macrocycle. The mucoadhesive and thermosensitive properties of calix[4]resorcinol–haloperidol nanoparticles were established by UV-, fluorescence and CD spectroscopy data. Pharmacological studies have revealed low in vivo toxicity of pure calix[4]resorcinol (LD_50_ is 540 ± 75 mg/kg for mice and 510 ± 63 mg/kg for rats) and the absence of its effect on the motor activity and psycho-emotional state of mice, which opens up a possibility for its use in the design of effective drug delivery systems. Haloperidol formulated with calix[4]resorcinol exhibits a cataleptogenic effect in rats both when administered intranasally and intraperitoneally. The effect of the intranasal administration of haloperidol with macrocycle in the first 120 min is comparable to the effect of commercial haloperidol, but the duration of catalepsy was shorter by 2.9 and 2.3 times (*p* < 0.05) at 180 and 240 min, respectively, than that of the control. There was a statistically significant reduction in the cataleptogenic activity at 10 and 30 min after the intraperitoneal injection of haloperidol with calix[4]resorcinol, then there was an increase in the activity by 1.8 times (*p* < 0.05) at 60 min, and after 120, 180 and 240 min the effect of this haloperidol formulation was at the level of the control sample.

## 1. Introduction

Haloperidol is known as a psychotropic drug with antipsychotic, neuroleptic, and antiemetic activities. This drug has low water solubility and high permeability through biological tissues [1,2]. Due to the fact that the low water solubility of haloperidol limits its therapeutic potential, the effect of its action with various routes of administration is currently being widely studied. Nasal administration is considered as a promising approach to deliver drugs to the brain [3]. However, with intranasal administration of haloperidol, its bioavailability is limited since the drug must be in solution for its effective absorption. Dissolution becomes the rate-limiting step if it occurs more slowly than absorption. Varying the composition of the dosage form can change the rate of dissolution and thus control the overall absorption. Different approaches to overcome poor aqueous solubility of drugs, such as solid dispersion and liquisolid dispersion techniques [4,5], micellar solubilization [6,7], an inclusion of an active poorly water-soluble drug in various nanoparticles [8,9,10], cyclodextrins [11,12,13,14], and liposomes [15,16] are often used to increase a drug’s solubility and, as a result, improve its bioavailability. In addition, there are several studies describing the inclusion of water-insoluble drugs into a calixarene cavity to form guest-host complexes [17,18,19,20], with a significant improvement of the in vitro dissolution profile and, in some cases, an increase in bioavailability [18,20] and a decrease of in vivo acute oral toxicity compared to the pure drug [20].

There are known examples of incorporating haloperidol into formulations based on lipids, surfactants and their mixtures [21], for the intranasal delivery of haloperidol, with improved pharmacokinetic profiles compared to the individual drug. In addition, the loading of haloperidol into nanoparticles based on anionic PEGylated Eudragit^®^ L100-55 and cationic Eudragit^®^ EPO polymers significantly increased the in vivo nose-to-brain delivery of haloperidol [22]. However, when haloperidol was loaded into hybrid nanoparticles based on the Eudragit L100-55 copolymer and the polymeric surfactant Brij98, its absorption was slowed down, which was expressed by a decrease in the pharmacological activity (cataleptogenic effect) compared to its free form [23]. Attempts to develop intranasal formulations are caused by problems with the oral delivery of haloperidol, when the drug enters systemic circulation after undergoing first-pass metabolism. In addition, haloperidol is 90% bound to plasma proteins, which leads to its low oral availability. Meanwhile, the intranasal administration of haloperidol can provide targeted delivery to the brain, bypassing the blood-brain barrier.

The nasal mucosa also acts as a barrier to drug penetration through the epithelial cells. The use of particles capable of adhering to mucous membranes as carriers may facilitate the retention of drugs on the surface providing more time for their absorption and increasing the effectiveness of their action. In our recent work, using porcine gastric mucin (PGM) as a model mucin, we have shown that viologen derivatives of calix[4]resorcinol interact with mucin strongly and, in combination with a drug (caffeine), the mixed systems exhibit mucoadhesive properties [24]. Due to their amphiphilic structure, the studied calixarenes are able to spontaneously self-assemble in aqueous solution with the formation of large aggregates, in which sparingly soluble drugs can be solubilized. In addition, π-electron-deficient 4,4′-bipyridinium units at the upper rim of the macrocyclic platform are able to selectively bind π-electron-rich molecules to form guest-host complexes [25], and also provide aqueous solubility, which is an important requirement for the development of efficient drug delivery carriers. Moreover, the search for new types of carriers of hydrophobic drugs, including haloperidol, remains an urgent task, since the previously studied formulations based on lipids, surfactants, their mixtures, and polymers are not without drawbacks, such as, for example, a laborious and multi-stage procedure for preparing nanoparticles using organic solvents and temperature, which does not meet the criteria of green chemistry. The formation of a complex between an amphiphilic calixarene and a hydrophobic drug can not only increase the aqueous solubility of the latter, but leads to the spontaneous formation of nanosized particles, which can make such macrocycles more preferable for formulating lipophilic biologically active substances. It should be noted that, to the best of our knowledge, there are no studies on the incorporation of haloperidol into calixarene molecules in the literature. The aim of this work was to prepare nanoparticles based on calix[4]resorcinol bearing viologen groups on the upper rim and decyl chains on the lower rim (**VC10**) and haloperidol (**Hal**) (Figure 1), to study the in vivo toxicity of **VC10** and its neurotropic properties, and to evaluate the cataleptogenic effect of the **VC10**–**Hal** formulation in comparison with a commercial formulation of **Hal**.

## 2. Materials and Methods

### 2.1. Chemicals and Reagents

Calix[4]resorcinol **VC10** was synthesized by the reaction of tetra-(bromomethyl)calix[4]resorcinol with monomethylviologen using the procedure described in the literature [26]. Haloperidol, **Hal** (Alfa Aesar), D_2_O (99.9 atom% D, Carl Roth GmbH), mucin from porcine stomach, and PGM (Type III, bound sialic acid 0.5–1.5%, partially purified powder, Sigma-Aldrich (St. Louis, MO, USA)) were purchased and used without further purification. Deionized, ultrapure water with a resistivity of 18.2 MΩ was generated using a Direct Q-5 UV water purification system from Millipore SAS (Molsheim, France) and was used throughout this work.

### 2.2. Methods

#### 2.2.1. UV-Vis Spectroscopy

The UV-Vis absorption spectra were recorded using a Specord 250 Plus spectrophotometer (Analytik Jena AG, Jena, Germany) equipped with a temperature controller accessory (Peltier). Measurements were carried out using a 1-mm quartz cuvette at 25 °C, over a wavelength range of 220–500 nm. The turbidity (absorption, A) of the samples in turbidimetric titration was measured at λ = 500 nm in a 1-cm quartz cuvette.

#### 2.2.2. NMR Spectroscopy

The ^1^H NMR spectra were recorded using a Bruker AVANCE(III)-600 spectrometer (Rheinstetten, Germany) operating at 600.1 MHz, which was equipped with a 5 mm broadband inverse probe head with z-gradient accessories to produce a field gradient up to 50 G·cm^–1^. The samples were prepared in D_2_O. The spectra were recorded at 303.0 ± 0.2 K. The chemical shifts are reported in the ppm scale and refer to the solvent (δ(HDO) 4.7 ppm).

#### 2.2.3. Dynamic Light Scattering

The hydrodynamic diameters of the **VC10**–**Hal** systems (molar ratio 1:1) at different concentrations (0.025 mM, 0.05 mM, 0.075 mM, 0.1 mM, 0.15 mM, 0.2 mM, 0.25 mM, 0.3 mM) were measured via the dynamic light scattering method using a Malvern Zetasizer Nano ZS particle size analyzer (Malvern Instruments Ltd., Worcestershire, UK). The light source was a 4 mW He-Ne laser with a wavelength of 632.8 nm. All measurements were carried out in disposable polystyrene cells in automatic mode at 25 °C, at a fixed scattering angle of 173°. The mean particle size ± standard deviation from three measurements was calculated using Malvern Zetasizer Software.

#### 2.2.4. Transmission Electron Microscopy

The morphology of the **VC10**–**Hal** aggregates was examined using electron microscopy with a Hitachi HT7800 transmission electron microscope (Hitachi High-Tech Science Corporation, Tokyo, Japan). The sample for imaging was prepared in deionized water; 5 μL of the solution was applied straight onto a 3.05 mm diameter copper grid with a formvar film (01700-F, Ted Pella, Inc., Redding, CA, USA) and dried at room temperature. The grid with the dried sample was placed in the transmission electron microscope using a special holder, followed by imaging at an accelerating voltage of 80 kV in the TEM mode.

#### 2.2.5. Fluorescence Spectroscopy

The fluorescence spectra were recorded by a Hitachi F-7100 Fluorescence Instrument (Hitachi High-Tech Science Corporation, Tokyo, Japan) with a xenon lamp as an excitation source and a temperature controller unit. Steady state fluorescence spectra were recorded using a 1 cm path length quartz cuvette (Hellma Analytics, Müllheim, Germany) in the range from 290 to 450 nm with an excitation light wavelength of 270 nm at temperatures of 298 K, 305 K, and 312 K. The slit widths of excitation and emission were set as 5 nm and 5 nm, respectively. The emission spectra of PGM at a constant concentration (0.05 mg/mL) were recorded in a titration series with a gradual increase in the concentration of **VC10** and **VC10**–**Hal** (molar ratio 1:1). The fluorescence intensity, at a wavelength of 330 nm, was used for the calculations given in Section 3.3.

#### 2.2.6. Circular Dichroism Spectroscopy

Circular dichroism (CD) spectra were recorded on a JASCO J-1500 spectropolarimeter (JASCO Corporation, Tokyo, Japan). Measurements were carried out using a 1 mm path length quartz cuvette at room temperature (25 °C). CD spectra were recorded from 250 to 190 nm with a bandwidth of 1 nm, scanning speed of 100 nm/min, and data pitch of 1 nm. In all samples, the concentration of PGM was 1 mg/mL. CD spectra were the average of three scans in each of two independent measurements.

#### 2.2.7. In Vivo Experiments

In vivo experiments were carried out using 62 male Wistar rats weighing 250–270 g and 54 white male mice weighing 18–22 g. Prior to the experiments, all animals were kept under standard vivarium conditions with a natural light regime and on a complete balanced diet in compliance with the International Recommendations of the European Convention for the Protection of Vertebrate Animals used in Experimental Research (1997), and the Rules of Laboratory Practice approved by the order of the Ministry of Health of the Russian Federation No. 199n, 1 April 2016. All in vivo experiments reported in this work were approved by the Ethical Committee of Kazan State Medical University (Protocol No. 8 of 30 October 2018).

##### Toxicity Assay

To determine the acute toxicity, a freshly prepared aqueous solution of **VC10** was administered intraperitoneally to male mice and male Wistar rats. Each study dose was applied to six animals. The dose that did not cause the death of any animal out of 6 (maximum tolerable dose) and the dose that caused the death of all 6 animals in the group (absolute lethal dose, LD_100_) were determined. The intermediate doses were also determined, the introduction of which led to the death of some animals. The experimental data were processed according to the method described by Behrens; the average lethal dose (LD_50_, the dose that causes the death of 50% of the animals) was determined graphically, and its error was determined using the Gaddam equation [27].

##### “Open Field” Test

The “Open Field” test is used to study the motor-exploratory activity of animals [28], the change in which reflects the effect of substances on the central nervous system (CNS). The experiment on mice was carried out in a round chamber with opaque walls, with a diameter of 97 cm and a wall height of 42 cm (Open Science, Russia). The installation floor was divided into sectors for the convenience of visual registration. In addition, there were holes in the floor imitating minks (2 cm in diameter). The criterion for the transition from one sector to another was the location of the hind legs of the animal in the new sector. After testing each mouse, the deodorization of the installation surface using 3% hydrogen peroxide was carried out. After each group of mice, the installation was treated with a 70% ethanol solution. In these experiments, each animal was placed in the center of the field and within 3 min of testing the following parameters were recorded: (1) the number of crossed lines, which reflects the nonspecific level of excitation (motor activity); (2) the number of holes examined, which is an indicator of research activity [29]; (3) the number of entrances to the center, which reflects the psycho-emotional state.

Calix[4]resorcinol **VC10** was administered intraperitoneally to 8 mice at doses of 5 mg/kg of mice weight (1/100 of LD_50_) and 10 mg/kg of mice weight (1/50 of LD_50_), 30 min prior to testing. Control animals were injected with an equivalent volume of saline.

Behavioral changes were recorded using EthoVision XT video tracking software for automatic track analysis by Noldus (Netherlands). This provided a more accurate quantification of the behavioral differences in the animals.

##### Catalepsy Test

In order to assess the effect of the **VC10**–**Hal** formulation, the method of cataleptogenic effect of haloperidol in rats was used, which reflects the pharmacological properties of neuroleptics [30]. The severity of catalepsy was assessed by the duration of the “lecturer’s posture” [31] using a special catalepsy device (Open Science, Russia), according to the protocol described in ref. [32]. Two series of experiments were carried out. In the first series, an aqueous solution of the binary **VC10**–**Hal** system was administered intranasally to 8 rats; in the second series it was administered intraperitoneally to 8 rats (the dose of haloperidol in all samples was 0.5 mg/kg of rat weight; the dose of **VC10** was 0.0028 mg/kg of rat’s weight). A commercial haloperidol formulation (5 mg/mL of a sterile solution containing lactic acid (Gedeon-Richter, Budapest, Hungary)) was used as a control.

In the first series of experiments, the **VC10**–**Hal** formulation was instilled into the nostrils of rats for 5 min using a specially designed plastic cannula, and then each rat was placed in a special catalepsy device with a plastic bar at a height of 10 cm. Each rat was carefully placed on a plastic bar at 5, 10, 20, 30, 60, 120 and 180 min after the intranasal and intraperitoneal administration of the **VC10**–**Hal** formulation, or the commercial formulation of haloperidol as a control, and the ability to maintain the “lecturer’s posture” (the duration on the bar) for 180 s was recorded. If the time spent on the bar reached 180 s, the rat was carefully removed. After each experiment, the rats were immediately returned to their cages. In the second series of experiments, intraperitoneal administration was carried out at the same dose and concentration as in the case of the intranasal administration.

##### Statistical Analysis

The obtained results were processed statistically according to the standard method using the two-sample Student’s *t*-test, after checking the data for the normal distribution in the compared groups, as well as the equality of the general variances in the GraphPad Prism 8.0.1 program. The results of the behavioral tests are presented as the mean ± standard deviation. The critical level of significance was set at *p* < 0.05.

## 3. Results and Discussions

### 3.1. Coaggregation of Haloperidol and Viologen Decyl Calix[4]resorcinol

The choice of calix[4]resorcinol **VC10** as a solubilizer of haloperidol molecules was based on our previous studies, where it was shown that (1) the macrocycle has low cytotoxicity (IC_50_ > 500) on the normal Chang liver cell line [33]; (2) has a low aggregation threshold of 0.3 mM in aqueous solutions and, due to its amphiphilic nature, is capable of solubilizing hydrophobic molecules into their aggregates [34]; (3) exhibits pronounced mucoadhesive properties both individually and in combination with drug molecules [24].

As mentioned above, **Hal** is practically insoluble in water. The UV-Vis absorption spectrum of **Hal** in ethanol shows that, at a concentration of 0.15 mM, it has limited absorption in the UV and visible region (Figure 2). With equimolar mixing of **VC10** with **Hal** after 15 min of sonication and 2 h of stirring at room temperature, the latter, visually, completely dissolves in aqueous solution due to solubilization by calix[4]resorcinol aggregates. The UV-Vis absorption spectrum of **VC10** in water shows a characteristic absorption peak at 260 nm (Figure 2). In the spectrum of the binary **VC10**–**Hal** system, there is an increase in the intensity of this peak and a slight hypsochromic shift by 3 nm. The maximum increase in absorbance is achieved after 24 h of mixing **VC10**–**Hal** at room temperature. The observed hyper- and hypsochromic effects in the UV-Vis absorption spectra indicate **Hal** binding in the hydrophobic domain of the **VC10** aggregates.

An equimolar mixture of **VC10**–**Hal** was studied by NMR spectroscopy. In the ^1^H-NMR spectrum of the binary **VC10**–**Hal** system, a new set of signals appears in the region of 6.85 and 7.71 ppm, as well as a broad peak in a strong field (2.39–2.90 ppm) in comparison with the spectrum of the pure **VC10** (Figure 3). In addition, an increase in the integral intensity is observed in the region of the H8 aromatic protons and H10 methylene protons of **VC10**, since in these regions there is an overlap with the signals of the aromatic (H8ʹ) and aliphatic (H4′, H7′ H10′) protons of **Hal**, respectively. The appearance of the signals of hydrophobic **Hal** in D_2_O confirms its solubilization by amphiphilic **VC10** aggregates.

According to the DLS data, the **VC10**–**Hal** aggregates are formed at concentrations an order of magnitude lower than the critical aggregation concentration (CAC) of individual **VC10** equal to 0.3 mM [34]. The addition of **Hal** promotes aggregation of the amphiphilic **VC10**. It was shown that the equimolar addition of **Hal** to aqueous solutions of **VC10** with concentrations up to CAC leads to the formation of larger particles with an average diameter from ~127 to ~198 nm depending on the concentration of **VC10**, with a monomodal size distribution and a high-quality correlation function (Figure 4, Appendix A). This behavior is consistent with observations for conventional or polymeric surfactants, where the addition of apolar solutes promotes the aggregation of surfactant molecules, which otherwise tend to disperse in the aqueous phase [35].

### 3.2. Morphology of Aggregates Based on Haloperidol and Viologen Decyl Calix[4]resorcinol

The TEM images of the binary **VC10**–**Hal** system clearly show the presence of spherical vesicles, which are presented in Figure 5a as dark spherical particles; light spots do not have a clear spherical shape, it is a feature of sample drying during sample preparation. The aggregates based on **VC10**–**Hal** are uniformly distributed over the surface of the copper grid with the predominance of particles 100 nm in size (Figure 5b). It should be noted that micellar aggregates with a diameter of 10–50 nm were observed in the TEM image of pure **VC10** [33]. Binding of **Hal** by calix[4]resorcinol **VC10** leads to a significant change in the morphology of aggregates in aqueous solution with the transition of micelles to vesicles. Apparently, the solubilization of **Hal** molecules disrupts the high curvature of the surface of micellar aggregates, which leads to the formation of membrane structures. Morphological changes are typical during the solubilization of hydrophobic drugs by micellar systems. Thus, solubilization of 5-methyl salicylate within micelles of copolymers of polyoxyethylene and polyoxypropylene significantly enlarges and changes the shape of the micellar structures from spherical to ellipsoidal [36]. The localization of the drug in the hydrophobic micellar core promotes the formation of higher morphologies. In another work, hydrophobic perphenazine solubilized into taurocholate/lecithin aggregates caused a change in morphology from spherical particles to wormlike micelles [37]. Summarizing the results obtained by UV spectrophotometry, NMR spectroscopy, DLS and TEM, it can be concluded that haloperidol is located in the hydrophobic region within the bilayer formed by the non-polar alkyl tails of calix[4]resorcinol.

### 3.3. Mucoadhesive Properties of Calix[4]resorcinol–Haloperidol Nanoparticles

To assess the mucoadhesive properties of calix[4]resorcinol–haloperidol nanoparticles, turbidimetric titration of PGM with a solution of binary **VC10**-**Hal** (1:1) system was carried out according to the procedure described in the literature [38]. The turbidity of solutions at various ratios was estimated from UV spectra at 500 nm, since at this wavelength there is no absorption in the UV spectra of **VC10** and **Hal**. The turbidimetric titration plot (Figure 6) shows that **VC10**–**Hal** aggregates have a good affinity for mucin due to the electrostatic interaction of positively charged viologen groups with negatively charged fragments in mucin. With an increase in the mass ratio of **VC10**–**Hal**, aggregation of mucin particles is observed with a maximum turbidity (absorption value A~1.4) at [**VC10**–**Hal**]/[mucin]~0.25, and with a further increase in the mass ratio, a decrease in turbidity and a disaggregation are observed. A similar profile was observed for the titration of mucin by individual **VC10** with a maximum absorption value of A~1.2 at 500 nm in our recent study [24]. A comparison of the turbidimetric titration curves of mucin by **VC10** and **VC10**–**Hal** shows that, despite the coaggregation of **VC10** and **Hal**, the ability of the macrocycle to effectively interact with mucin is maintained. Moreover, it should be noted that the maximum absorption on the turbidimetric curve for the **VC10**–**Hal** system is 0.2 units higher than that for the pure **VC10**, which indicates a more efficient interaction of **VC10** with mucin in the presence of **Hal**. Probably, the reason for the enhanced mucoadhesive effect is the vesicular form of the **VC10**–**Hal** aggregates, which has a lower surface curvature compared to the micellar structures of individual **VC10**.

The binding of **VC10**–**Hal** aggregates to mucin was assessed from fluorescence quenching plots. It is known that mucin has intrinsic fluorescence due to fluorescent chromophores, namely amino acid residues, mainly tryptophan in the protein part of the molecule. In an aqueous solution, the excitation spectrum of tryptophan is in the wavelength range from 200 to 300 nm, and the fluorescence spectrum is recorded from 300 to 440 nm. The intensity of tryptophan fluorescence is highly dependent on the microenvironment, so the study of this fluorescence provides valuable information when binding proteins to various molecules. In our recent work, it was shown that the addition of **VC10** results in quenching of the mucin fluorescence intensity at 298 K [24]. In this work, we performed a series of fluorometric titrations of mucin with pure **VC10** and **VC10**–**Hal** at different temperatures (298 K, 304 K, 310 K) in order to determine the quenching mechanism. The concentration of mucin and calixarene was chosen so that the absorption value in their UV spectra did not exceed 0.1 unit. This minimizes the effect of the internal filter on the fluorescence spectra. With the gradual addition of **VC10** and **VC10**–**Hal** (at a ratio of 1:1) to PGM at a constant concentration (0.05 mg/mL) at different temperatures, the fluorescence of the latter is intensely quenched (Appendix A). These spectral changes clearly indicate the presence of specific interactions of **VC10** and **VC10**–**Hal** with PGM. The mechanism of the observed fluorescent quenching can be dynamic or static, or a combination of both of these processes. In general, the Stern–Volmer Equation (1) is used to reveal the quenching mechanism:F_0_/F = 1 + K_sv_·[Q],(1)
where F_0_ и F are the fluorescence intensities of PGM in the absence and in the presence of the quencher Q (**VC10**–**Hal** and **VC10**); K_sv_ is the Stern–Volmer quenching constant; and Q is the concentration of the quencher (**VC10**–**Hal** and **VC10**).

Stern–Volmer plots for the fluorescence quenching of PGM by **VC10** and **VC10**–**Hal** at 298 K, 304 K and 310 K, are shown in Appendix A. The graphs have the *y*-intercept equal to one, and the slope equal to K_sv_. In the case of one type of quenching, the graph in the Stern–Volmer coordinates is strictly linear, which is what we observe for our systems. The plots in Appendix A showed a good linear relationship at every experimental temperature within the studied concentration range. From the obtained data (Table 1), it can be seen that the quenching constants K_sv_ of PGM in the presence of the quencher (**VC10** and **VC10**–**Hal**) decrease with increasing temperature, which indicates static quenching [39]. When static quenching occurs, the fluorescent molecule forms a non-fluorescent complex with the quencher. As the quencher concentration increases, less unbound fluorophore remains in the solution and the fluorescence intensity decreases.

Further, to determine the binding constant (K_a_), the number of binding sites (*n*), and the dissociation constants (K_d_) for the PGM–**VC10** and PGM–[**VC10**–**Hal**] systems, the double logarithm regression curves as a function of quencher concentration were plotted based on Equation (2):log((F_0_ − F)/F) = logK_a_ + nlog(Q).(2)

In the graph, *n* is equal to the tangent of the slope, and K_a_ is the point of intersection with the y-axis. The dissociation constant (K_d_) is the reciprocal of the association constant, K_d_ = 1/K_a_. Table 2 summarizes the calculated data of *n*, K_a_, K_d_.

It can be seen from Table 2 that the *n* value for both systems at different temperatures is close to 1, which indicates one binding site in the PGM–**VC10** and PGM–[**VC10**–**Hal**] systems. The probable site of PGM binding to the positively charged viologen calix[4]arene **VC10** is the sialic acid carboxyl groups in mucin. Considering that the binding constants (logK_a_) are in the range of 6.45–7.96, it can be assumed that **VC10** and **VC10**–**Hal** have a strong affinity for mucin at different temperatures. The strongest ability to bind with mucin is observed at 298 K (logK_a_ = 7.47 for PGM–**VC10** and logK_a_ = 7.96 for PGM–[**VC10**–**Hal**]). With increasing temperature, the values of the binding constants in both systems decrease, which additionally points to a static quenching mechanism. As the temperature rises, the intramolecular mobility increases, the stability of the formed complexes decreases, and, accordingly, the values of the static quenching constants decrease. It should be noted that for the PGM–[**VC10**–**Hal**] system, as the temperature rises to physiological, the binding constant decreases by 1.5 units, while the decrease occurs by 0.6 units in the PGM–**VC10** system. When comparing both systems, it can be concluded that the ternary system in the presence of a drug is more thermosensitive despite the greater stability of vesicular structures compared to micelles, which is an advantage in the development of a drug delivery system. A likely cause of thermosensitivity, in the case of using the binary system **VC10**–Hal, may be a destructive change in mucin morphology as a result of a stronger interaction in the PGM–[**VC10**–**Hal**] system compared to the PGM–**VC10** system. A possible explanation for this effect is the lower curvature of the vesicle surface, which ensures interaction with a larger surface of mucin compared to micellar particles. It should be additionally emphasized that, similar to micellar aggregates, the vesicles spontaneously formed in the binary system **VC10–Hal** are dynamic assemblies that can be subjected to structural rearrangements under different stimuli including temperature induced changes [40,41].

Using K_a_ from Table 2, the Gibbs free energy variation (ΔGº) was calculated using Equation (3):ΔGº = −RT·lnK_a_, (3)
where R is the gas constant (8.314 Jmol^−1^K^−1^), and T are the experimental temperatures (298 K, 304 K, 310 K). The calculated values of ΔG° are summarized in Table 3. The negative sign for ΔGº indicates that the binding of PGM with **VC10** and **VC10**–**Hal** is a spontaneous process.

To determine the conformational stability of PGM upon its interaction with **VC10** and binary **VC10**–**Hal** system, circular dichroism (CD) spectra were recorded at a constant mucin concentration and a variable concentration of the studied systems (Figure 7). The CD spectrum of the individual PGM has a wide band in the far UV region, with a minimum at 206 nm and a negative ellipticity value, which indicates its secondary structure (random coil), and which is consistent with the literature data [42]. The signal is due to amino acid residues in the mucin having -NHCO- chromophores, which absorb in the 200–210 nm region. In the binary PGM–**VC10** and ternary PGM–[**VC10**–**Hal**] systems, the spectrum profile is typical for the pure PGM, which indicates that the random coil conformation of mucin is retained upon its interaction with the calix[4]arene **VC10** and **VC10**–**Hal** nanoparticles. With an increase in the proportion of the system (**VC10** and **VC10**–**Hal**) added to PGM, the value of negative ellipticity at the minimum decreases, and this minimum shifts to 209 nm, which is caused by a rather strong interaction in the binary and ternary systems, which was confirmed above by the UV and fluorescence spectroscopy data. The increase in the **VC10** concentration led to changes in the amplitude of the spectrum, indicating that the specific binding of the viologen groups of **VC10** to carboxylate groups in the glycoprotein increases the disorder of the secondary structure due to the inclusion of macrocycle molecules in the peptide backbone [43].

### 3.4. Toxicity Studies

To identify the toxic and safe doses of **VC10** in order to use it as a carrier/delivery system, a study of acute toxicity and the effect of **VC10** on indicators in the “Open Field” in rodents (to assess the level of emotional and behavioral activity of animals) was carried out. The results of the study of acute toxicity of **VC10** when administered intraperitoneally allowed the calculation of the LD_50_ values in mice and rats, which served as the basis for determining the working doses in further experiments. The LD_50_ values of **VC10** were found to be 540 ± 75 mg/kg for mice and 510 ± 63 mg/kg for rats, which allows calix[4]resorcinol **VC10** to be classified as a toxicity category 4, i.e., as a compound with low toxicity [44].

The behavior in the “Open Field” indicates the functional state of the CNS of mice, which is the main marker of the toxic effects of any substances on the body. The “Open Field” test is used to study the behavior of rodents in new conditions and allows researchers to evaluate the dynamics of individual behavioral elements. This test creates a mild anxiety model. The calix[4]resorcinol **VC10** was administered intraperitoneally at doses of 1/100 and 1/50 of the LD_50_ 30 min before the experiment. Mice from the control group received the appropriate volumes of saline. Appendix A shows that the average number of examined holes in the studied groups, taking into account the experimental error, is comparable to the control group, where the mice were not injected with the test compound. The maintenance of exploratory activity in the mice suggests that the cognitive functions of animals are not impaired by **VC10**. The number of crossed lines in all three groups was approximately the same (Appendix A), which indicates the absence of an effect of **VC10** on the motor activity of mice. In the absence of any shelters, the animal feels more secure being near the wall. The number of entrances to the center did not change for the animals that received **VC10** in a dose of 1/50 of the LD_50_, compared with the control group (Appendix A). This means that calixarene does not have a depressing effect on the psycho-emotional status of mice. The primary pharmacological tests revealed the low toxicity of **VC10** and the absence of its effect upon behavioral characteristics at the studied doses, on the basis of which it can be concluded that **VC10**, when administered once, does not affect the CNS of laboratory mice. The obtained results gave us the basis to carry out further in vivo experiments on the study of the cataleptogenic effect of haloperidol in the formulation with **VC10**.

### 3.5. In Vivo Catalepsy Tests

Catalepsy tests in rats were performed to estimate the efficacy of the **VC10**–**Hal** formulation compared to the commercial formulation of **Hal**. Cataleptogenic effect (catalepsy), i.e., the loss of the ability to make voluntary movements and the ability to maintain an artificially uncomfortable position for a long time, is one of the manifestations of extrapyramidal symptoms as a side effects of drugs with neuroleptic activity [45]. The study of the cataleptogenic action of **Hal** was carried out in two series of experiments, namely, with the intranasal and intraperitoneal administration of the studied formulations.

Intranasal administration for the treatment of diseases of the CNS is attractive because it allows the drug to be delivered directly to the brain, bypassing the blood-brain barrier. At the same time, neurotherapeutic drugs are delivered to the brain in significant concentrations with minimal exposure to systemic circulation [46]. The results of the cataleptogenic effect of **Hal** in the formulation with **VC10,** and the commercial formulation of **Hal** (as a control), with intranasal administration are shown in Figure 8. Analysis of intranasal administration showed the severity of the cataleptogenic effect of **Hal** in the formulation of **VC10**–**Hal** in the first 120 min of the study, which is comparable to the effect of commercial **Hal**. However, at 180 and 240 min, the catalepsy duration of the test sample was lower than in the control one.

As for the analysis of the intraperitoneal administration of the test sample in comparison with the commercial **Hal**, it should be noted that at 10 and 30 min there was a significant decrease in the intensity of the cataleptogenic effect of **Hal** in the formulation with **VC10**, and at 60 min, on the contrary, an increase in the effect (Figure 9). Further, at 120, 180 and 240 min, the effect of the formulated haloperidol was at the level of the control sample. The lower activity in the first 30 min may be due to the delayed release of **Hal** from the **VC10**–**Hal** formulation. The results of the cataleptogenic effect of **Hal** with intranasal and intraperitoneal administration of the studied samples showed that, depending on the route of administration, different results are observed over time, which, apparently, is due to the difference in pharmacokinetic parameters.

## 4. Conclusions

A water-soluble formulation of **Hal** was obtained by solubilizing it in the aggregates of calix[4]resorcinol **VC10** to form vesicles. **Hal** was found to be located in the hydrophobic region within the bilayer formed by the non-polar alkyl tails of calix[4]resorcinol. Effective mucoadhesive properties of **VC10**–**Hal** nanoparticles were shown by UV, fluorescence, and CD spectroscopy. A decrease in the binding constant in the PGM–[**VC10**–**Hal**] system with an increase in temperature to physiological level provides the thermosensitive properties of **VC10**–**Hal** nanoparticles. According to CD spectroscopy data, the spectrum profile of the PGM–[**VC10**–**Hal**] system corresponds to the spectrum of pure PGM, which indicates the conformational stability (random coil conformation) of mucin upon interaction with **VC10**–**Hal** nanoparticles.

In vivo experiments revealed the low toxicity of **VC10** (LD_50_ is 540 ± 75 mg/kg for mice and 510 ± 63 mg/kg for rats), as well as the absence of its effect on motor activity and psycho-emotional status, which allows consideration of possibly using the binary **VC10**–**Hal** systems in further in vivo studies. The results of the experiments on the study of catalepsy in rats showed that with the various routes of drug administration (intranasal and intraperitoneal), **Hal** formulated with **VC10** exhibits a pronounced cataleptogenic effect comparable to commercial **Hal**. This work demonstrated the successful use of cationic calixarene as a carrier for a hydrophobic drug in vivo for the first time, which opens up a perspective for its application in the development of nanoscale systems for the delivery of poorly soluble drugs to the brain.

## Figures and Tables

**Figure 1 pharmaceutics-15-00921-f001:**
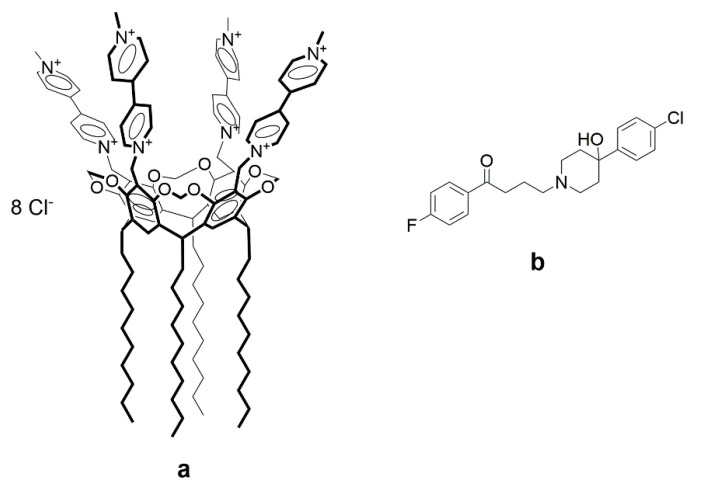
Molecular structures of calix[4]resorcinol **VC10** (**a**) and **Hal** (**b**).

**Figure 2 pharmaceutics-15-00921-f002:**
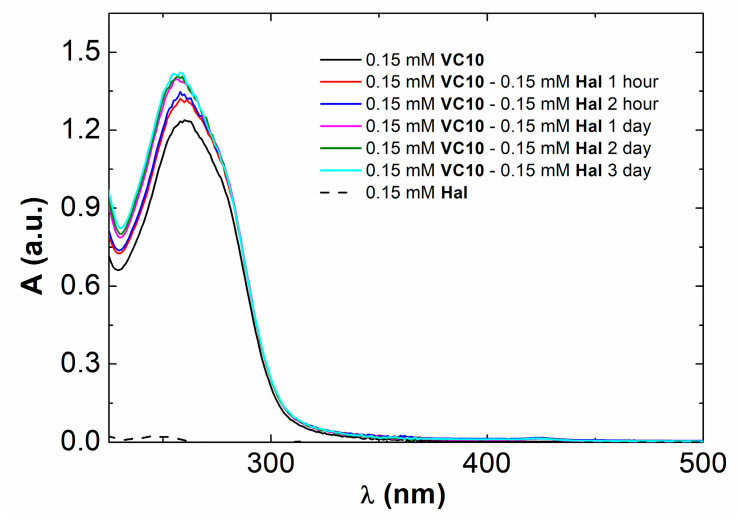
UV-Vis absorption spectra of 0.15 mM **VC10**, 0.15 mM **VC10**–0.15 mM **Hal** in H_2_O, and 0.15 mM **Hal** in EtOH.

**Figure 3 pharmaceutics-15-00921-f003:**
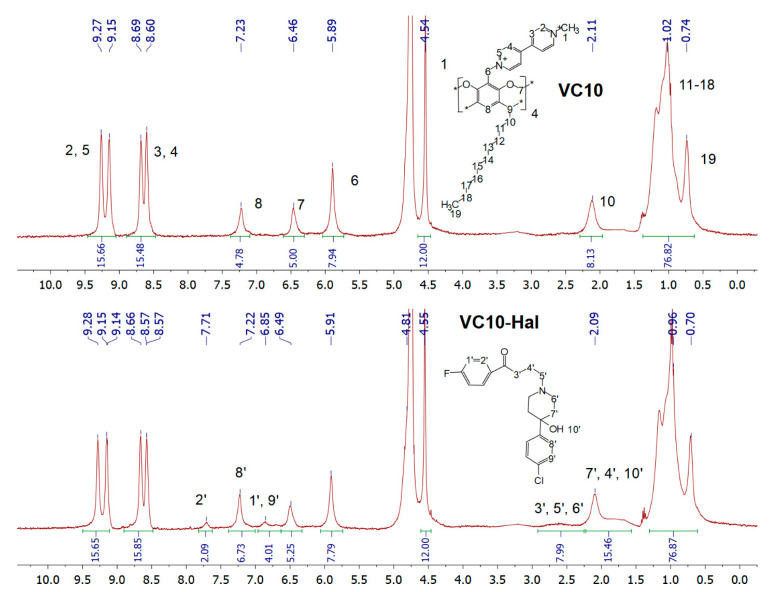
^1^H-NMR spectra of 2 mM **VC10** and the binary 2 mM **VC10**–2 mM **Hal** system in D_2_O.

**Figure 4 pharmaceutics-15-00921-f004:**
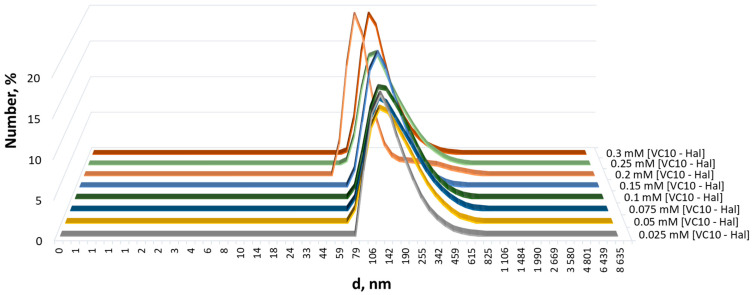
Particle size distribution by number in aqueous solutions of VC10–Hal (1:1) at different concentrations.

**Figure 5 pharmaceutics-15-00921-f005:**
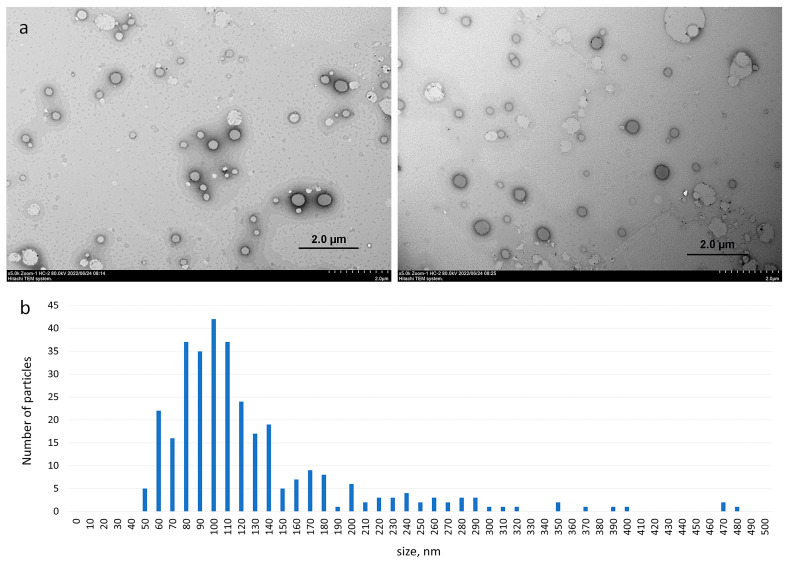
Transmission electron micrographs of **VC10**–**Hal** (**a**) and the histogram showing the average particle size obtained from the TEM images using ImageJ software (**b**).

**Figure 6 pharmaceutics-15-00921-f006:**
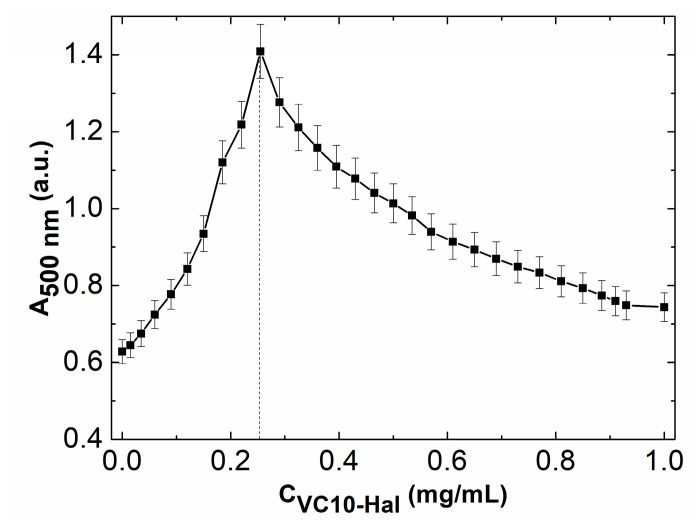
Turbidimetric titration of 1 mg/mL PGM with 1 mg/mL solution of **VC10**–**Hal** (1:1).

**Figure 7 pharmaceutics-15-00921-f007:**
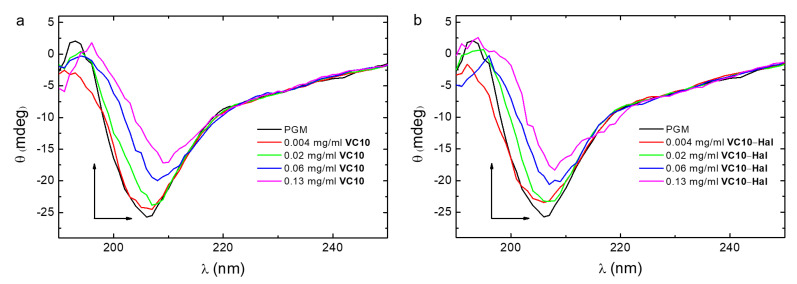
UV circular dichroism spectra of the PGM, PGM–**VC10** (**a**) and PGM–[**VC10**–**Hal**] (**b**) systems in aqueous solution. The concentration of PGM was constant (1 mg/mL) [**VC10**–**Hal**] = 1:1.

**Figure 8 pharmaceutics-15-00921-f008:**
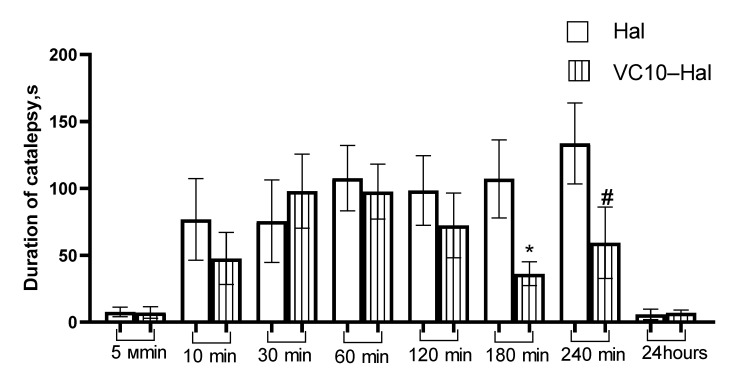
Duration of catalepsy after intranasal administration (s). * *p* < 0.05—Statistically significant difference compared to the haloperidol group. # 0.05 < *p* < 0.1—Significance trend compared to haloperidol group.

**Figure 9 pharmaceutics-15-00921-f009:**
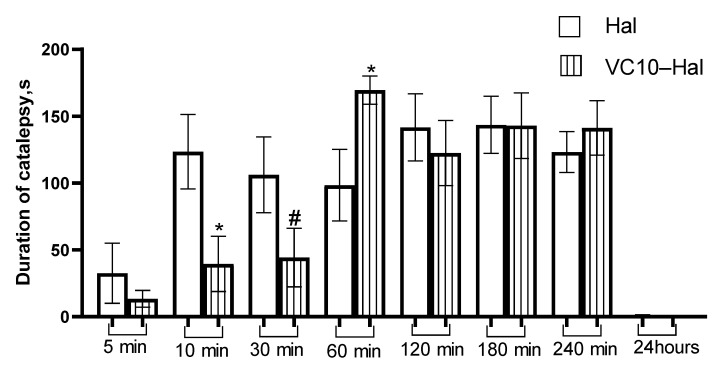
Duration of catalepsy after intraperitoneal administration (s). * *p* < 0.05—Statistically significant difference compared to the haloperidol group. # 0.05 < *p* < 0.1—Significance trend compared to haloperidol group.

**Table 1 pharmaceutics-15-00921-t001:** Quenching constants of **PGM** in the presence of the quencher (**VC10** and **VC10**–**Hal**) at different temperatures.

	K_sv_ 10^3^ (M^−1^)		
	298 K	304 K	310 K
PGM–**VC10**	1762 ± 28	1635 ± 42	1516 ± 42
PGM–**[VC10–Hal]**	1702 ± 39	1563 ± 39	1405 ± 25

**Table 2 pharmaceutics-15-00921-t002:** Number of binding sites (*n*), binding constants (K_a_), and dissociation constants (K_d_), for the PGM–**VC10** and PGM–[**VC10**–**Hal**] systems at different temperatures.

	PGM–VC10			PGM–[VC10–Hal]		
	*n*	logK_a_	K_d_ 10^−8^ (M)	*n*	logK_a_	K_d_ 10^−8^ (M)
298 K	1.21 ± 0.01	7.47 ± 0.06	3.40	1.30 ± 0.01	7.96 ± 0.04	1.10
304 K	1.17 ± 0.03	7.18 ± 0.17	6.28	1.12 ± 0.02	6.87 ± 0.11	13.25
310 K	1.13 ± 0.04	6.90 ± 0.29	11.06	1.05 ± 0.04	6.45 ± 0.22	32.71

**Table 3 pharmaceutics-15-00921-t003:** The Gibbs free energy variation (ΔGº) for the PGM–**VC10** and PGM–[**VC10**–**Hal**] systems at different temperatures.

	ΔG° (kJ·mol^−1^)		
	298 K	304 K	310 K
PGM–**VC10**	−42.62	−41.79	−40.95
PGM–[**VC10**–**Hal**]	−45.41	−39.98	−38.28

## Data Availability

Data are contained within this article.

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
