# Peer review of "Cataleptogenic Effect of Haloperidol Formulated in Water-Soluble Calixarene-Based Nanoparticles"

_pharmaceutics, 2023, doi:10.3390/pharmaceutics15030921_

Round 1
Reviewer 1 Report
In this manuscript, the authors reported interesting water-soluble calixarene-based nanoparticles, where the haloperidol formulated with viologen calix[4]resorcinol exhibits a pronounced cataleptogenic effect in rats both when administered intranasally and intraperitoneally. The data support the conclusions and I recommend its publication after the following comments are addressed.
1. For figure 5, there is white sphere-like particles and dark sphere-like particles. Please indicate which (white or dark) are the sample.
2. Stern-Volmer plots for fluorescence quenching of PGM by VC10 and VC10–Hal at 326 298 K, 304 K and 310 K. Please explain why these temperatures are chosen as the typical ones.
3. When comparing both systems, it can be concluded that the ternary system in the presence of a drug is more thermosensitive despite the greater stability of vesicular structures compared to micelles. Could the authors explain the reason?
4. For different approaches to overcome poor aqueous solubility of drugs, the recent approach literatures can be updated, DOI:10.1016/j.jconrel.2020.09.035, DOI:10.1039/d1nh00506e
5. For "which is caused by a rather strong interaction in binary and ternary systems, which was confirmed above by UV and fluorescence spectroscopy data." could the authors explain what possible interation are they?
Author Response
1. For figure 5, there is white sphere-like particles and dark sphere-like particles. Please indicate which (white or dark) are the sample.
Response: In Figure 5, the VC10–Hal is presented as dark spherical particles, while light spots do not have a clear spherical shape, it is a feature of sample drying during sample preparation. This point has been added in the revised manuscript.
2. Stern-Volmer plots for fluorescence quenching of PGM by VC10 and VC10–Hal at 326 298 K, 304 K and 310 K. Please explain why these temperatures are chosen as the typical ones.
Response: To determine the mechanism of fluorescence quenching, Stern-Volmer graphs are plotted at various temperatures. Temperature 298 K is usually the ambient temperature (room temperature). The maximum temperature of 310 K corresponds to the physiological temperature of the human body. Temperature 304 K is intermediate between them.
3. When comparing both systems, it can be concluded that the ternary system in the presence of a drug is more thermosensitive despite the greater stability of vesicular structures compared to micelles. Could the authors explain the reason?
Response: The binary system (VC10–Hal) is self-assembled into vesicles, and the micellar aggregates are formed in a solution of pure VC10. Thus, the interaction of mucin was studied both with vesicles based on VC10–Hal and micelles of VC10. A likely cause of thermosensitivity in the case of using the binary system VC10–Hal may be a destructive change in mucin morphology as a result of a stronger interaction in PGM–[VC10–Hal] system compared to PGM–VC10 system. A possible explanation for this effect is the lower curvature of the vesicle surface, which ensures interaction with a larger surface of mucin compared to micellar particles. It should be additionally emphasized that similar to micellar aggregates the vesicles spontaneously formed in the binary system (VC10–Hal) are dynamic assemblies that can be subjected to structural rearrangements under different stimuli including temperature induced changes [https://doi.org/10.1021/acsomega.1c00469; https://doi.org/10.1039/C5SM01825K]. This point has been added in the revised manuscript.
4. For different approaches to overcome poor aqueous solubility of drugs, the recent approach literatures can be updated, DOI:10.1016/j.jconrel.2020.09.035, DOI:10.1039/d1nh00506e
Response: These references have been included in the revised manuscript.
5. For "which is caused by a rather strong interaction in binary and ternary systems, which was confirmed above by UV and fluorescence spectroscopy data." could the authors explain what possible interation are they?
Response: The discussion of the change in the circular dichroism spectra was expanded with the following phrase: The increase of the VC10 concentration led to changes in the amplitude of the spectrum, indicating that the specific binding of the viologen groups of VC10 to carboxylate groups in the glycoprotein increases the disorder of the secondary structure due to the inclusion of macrocycle molecules in the peptide backbone [10.1080/07391102.2003.10506874].
Reviewer 2 Report
The submitted study presents the water-soluble form of haloperidol was obtained by coaggregation with viologen decyl calix [4] resorcinol to form vesicular nanoparticles. The formation of nanoparticles is achieved by spontaneous loading of haloperidol into the hydrophobic domains of aggregates based on calix[4]resorcinol. In addition, mucoadhesive and thermosensitive properties of calix[4]resorcinol–haloperidol nanoparticles were established by UV-, fluorescence and CD spectroscopy data. Literature review and importance of incorporating haloperidol into formulations based on lipids, surfactants and their mixtures for intranasal delivery of haloperidol with improved pharmacokinetic profiles compared to an individual drug is not sufficient, some gaps could not covered more comprehensively such as the haloperidol into nanoparticles based on anionic PEGylated Eudragit® L100-55 and cationic Eudragit® EPO polymers significantly increased the in vivo nose-to-brain delivery of haloperidol. In addition, the authors confirmed experimentally what was reported in the literature review in the introduction, its motivation need to be clarified. Hence, the correlation between novelty and aim of the study should be improved. Please also add some quantitative data in the abstract. It’s hard to understand the main results of the manuscript. I am encouraging and appreciate the authors to revise the manuscript according to the above suggestions that make it more interesting for the readers. I think it could be published with above mentioned major changes.
Author Response
Yes, indeed, the cataleptogenic effect of haloperidol is significantly higher when it is formulated with polyelectrolyte complexes. In our work, calixarene was used for the first time as a carrier of haloperidol in order to obtain its water-soluble form. It is worth noting that calixarenes have not previously been used for the formulation of haloperidol. This is the novelty of this work. In addition, the search for new formulations remains an urgent task, since previously studied formulations based on lipids, surfactants, their mixtures, and polymers are not without drawbacks, such as the multi-stage, time-consuming synthesis of nanoparticles using organic solvents. In our work, it was shown that interaction between calixarene and haloperidol leads to the spontaneous formation of vesicles (biomimetic nanoparticles) in aqueous solution under mild conditions without the use of high temperature and organic solvents, which meets the criteria of green chemistry. These points are added to the Introduction and Conclusions. Quantitative information on the in vivo toxicity of calixarene and the cataleptogenic effect of haloperidol formulated with macrocycle after intraperitoneal and intranasal administration has been added to the abstract.
Reviewer 3 Report
Comments
Quantitative information should be provided in the abstract section.
The author should mention in details of Figure 2 X-axis and Y-axis units.
Figure 5 is missing in the text. Also, the author should provide the histogram of the average particle.
Figures S2 –S4 missing and the Tables S1-S3 is missing in the main manuscript.
The conclusion section provided with outstanding point of this work.
Typographical errors must be corrected throughout the manuscript (i.e, superfluous spaces, inconsistent use of units, superscript, etc.).
Supporting Information file is missing in this submission.
Author Response
Quantitative information should be provided in the abstract section.
Response: Quantitative information on the in vivo toxicity of calixarene and the cataleptogenic effect of haloperidol formulated with macrocycle after intraperitoneal and intranasal administration has been added to the abstract of the revised manuscript.
The author should mention in details of Figure 2 X-axis and Y-axis units.
Response: Figure 2 shows the UV-Vis absorption spectra, which are a graphical representation of the dependence of the absorption intensity in arbitrary unit on the wavelength of the incident radiation in nanometers. In Figure 2, the X-axis unit has been added. The thickness of the cuvette was added in subsection 2.2.1.
Figure 5 is missing in the text. Also, the author should provide the histogram of the average particle.
Response: Reference to Figure 5 has been added to the text and the histogram of the average particle size has been added in Figure 5 of the revised manuscript.
Figures S2 –S4 missing and the Tables S1-S3 is missing in the main manuscript.
Response: Figures S2 –S4 and Tables S1-S3 are provided in the Supporting Information file, which has been added along with the revised manuscript.
The conclusion section provided with outstanding point of this work.
Response: This work demonstrated the successful use of cationic calixarene as a carrier for hydro-phobic drug in vivo for the first time, which opens up a perspective of its application in the development of nanoscale systems for the delivery of poorly soluble drugs to the brain. This point has been added in the revised manuscript.
Typographical errors must be corrected throughout the manuscript (i.e, superfluous spaces, inconsistent use of units, superscript, etc.).
Response: We apologize for typographical errors. We tried to eliminate them in the revision of the manuscript.
Supporting Information file is missing in this submission.
Response: We apologize for this shortcoming. Supporting Information file has been submitted along with the revised manuscript.
Round 2
Reviewer 2 Report
Accept in present form